# Informing antimicrobial stewardship with explainable AI

**Massimo Cavallaro**[1,2]*, **Ed Moran**[3], **Benjamin Collyer**[4], **Noel D. McCarthy**[5,6], **Christopher Green**[7,8], **Matt J. Keeling**[1,2]

**1** School of Life Sciences and Mathematics Institute, University of Warwick, Coventry, United Kingdom, **2** The Zeeman Institute for Systems Biology & Infectious Disease Epidemiology Research, University of Warwick, Coventry, United Kingdom, **3** Department of Infectious Disease, North Bristol NHS Trust, Bristol, United Kingdom, **4** Faculty of Medicine, School of Public Health, Imperial College London, London, United Kingdom, **5** Institute of Population Health, Trinity College Dublin, University of Dublin, Dublin, Ireland, **6** Warwick Medical School, University of Warwick, Coventry, United Kingdom, **7** Institute of Microbiology & Infection, University of Birmingham, Birmingham, United Kingdom, **8** Department of Infectious Diseases & Tropical Medicine, University Hospitals Birmingham NHS Foundation Trust, United Kingdom

* m.cavallaro@warwick.ac.uk

**Data Availability Statement:** Data contain sensitive and potentially identifying patient information if linked with other datasets, with public data deposition non-permissible. Interested readers wanting these data may wish to contact

## Abstract

The accuracy and flexibility of artificial intelligence (AI) systems often comes at the cost of a decreased ability to offer an intuitive explanation of their predictions. This hinders trust and discourage adoption of AI in healthcare, exacerbated by concerns over liabilities and risks to patients' health in case of misdiagnosis. Providing an explanation for a model's prediction is possible due to recent advances in the field of interpretable machine learning. We considered a data set of hospital admissions linked to records of antibiotic prescriptions and susceptibilities of bacterial isolates. An appropriately trained gradient boosted decision tree algorithm, supplemented by a Shapley explanation model, predicts the likely antimicrobial drug resistance, with the odds of resistance informed by characteristics of the patient, admission data, and historical drug treatments and culture test results. Applying this AI-based system, we found that it substantially reduces the risk of mismatched treatment compared with the observed prescriptions. The Shapley values provide an intuitive association between observations/data and outcomes; the associations identified are broadly consistent with expectations based on prior knowledge from health specialists. The results, and the ability to attribute confidence and explanations, support the wider adoption of AI in healthcare.

## Author summary

Antimicrobial resistance is the ability of organisms (usually bacteria) that cause infections to survive antibiotic treatments. It is a major threat to health and is responsible for an increased risk of death and prolonged hospital stays. Artificial intelligence (AI) is starting to be used for early prediction of resistance to different antibiotics, but care is needed to safely and confidently incorporate this tool into clinical practice. To gain trust from both patients and the medical profession, AI output needs to be transparent and explainable.

R&D@uhb.nhs.uk. All codes for data management and analysis are archived online at https://github.com/mcavallaro/ML4AMR.

**Funding:** This work was supported by Health Data Research UK, which is funded by the UK Medical Research Council, EPSRC, Economic and Social Research Council, Department of Health and Social Care (England), Chief Scientist Office of the Scottish Government Health and Social Care Directorates, Health and Social Care Research and Development Division (Welsh Government), Public Health Agency (Northern Ireland), British Heart Foundation and the Wellcome Trust supported MC, MJK, and NDM. MJK and NDM are affiliated to the National Institute for Health Research Health Protection Research Units (NIHR HPRUs) in Gastrointestinal Infections and in Genomics and Enabling Data. MJK is funded by UK Research and Innovation through the JUNIPER modelling consortium (MR/V038613/1). The funders had no role in study design, data collection and analysis, decision to publish, or preparation of the manuscript.

**Competing interests:** The authors have declared that no competing interests exist.

Here we use explainable AI to show how the characteristics of patients can be used to determine the chance of antimicrobial resistance. The identified patterns could potentially inform hospital practice. Our approach reports the level of certainty and uncertainty for each prediction. This can guide doctors on how much they should rely on it when making initial recommendations. We also show that following our AI predictions would have lowered the initial number of mismatched prescriptions compared to what happened in practice. These methods may therefore increase confidence in AI predictions, improve patient treatment and slow the increase in antimicrobial resistance by targeting antibiotics effectively.

## Introduction

As data becomes complex (and too large) for traditional statistics, artificial intelligence (AI) systems, that are able to generate predictions from such data, could form part of health decision making. This is particularly relevant in modern healthcare, given the abundance of information recorded and electronically available about each patient [1]. In addition to accelerate time-consuming decision processes, adoption of AI in healthcare also has the potential to reduce medical errors. However, despite their prospective benefits, there still are challenges to overcome in order to routinely incorporate AI systems into medical practice [2, 3]. In fact, compared to traditional statistical modelling, AI often comes at the cost of a decreased ability to heuristically explain the outputs and place any confidence in AI-derived information [4]. To underline the difficulties in illustrating how they yield a particular result, AI models have been often depicted as black-box systems [5]. This contrasts with the need for awareness of professionals in healthcare, where patients might ask: who is at fault if this black box malfunctions and health is at risk? In order to be able to use AI algorithms as tools, biomedical professionals should be able to inspect a black-box model, make sense of its outputs, and only then use its evidence in clinical practice. The public also considers transparency of AI systems more important in medical care than in other application domains [6]. This possibility is granted by the so-called explainable or interpretable AI [5, 7].

Here we are motivated by the challenge of predicting whether bacterial pathogens isolated from hospitalised patients are resistant to particular antibiotic drugs. At a public-health level, this issue of antimicrobial resistance threatens the efficacy of the available drugs over time, with antibiotic stewardship seen as critical for preserving a stockpile of effective treatments [8–10]. At the level of an individual patient, the efficacy of a treatment depends on matching antibiotic choice to the susceptibilities of the infecting pathogen. The susceptibilities can only be confirmed in laboratory tests [11, 12] but, to provide rapid interventions in clinical practice, antibiotic drugs are often empirically prescribed prior to receiving laboratory results, thus risking sub-optimal treatments [13].

In this paper, we present an analysis of a large population of hospital inpatients with Gram-negative bacteria isolated from blood and urine cultures. Several studies have demonstrated the ability of machine-learning algorithms such as the gradient boosted decision tree (GBDT) trained on admission data to predict the presence of antimicrobial resistance (AMR) in clinical settings [14–18]. Here, in addition to performing predictions and discussing their accuracy, we consider the GBDT-model predictions in more detail focusing on their predictive power for individual patients (reducing the number of prescriptions mismatched to resistant culture test results) as well as using Shapley values [7, 19–21] to unpick the underlying dependence of the model. These two factors help dispel the idea that machine-learning algorithms are just

black-boxes, providing health-care professionals with both a rationale and confidence levels for predicted patterns of antibiotic resistance.

## Materials and methods

This is a retrospective study of 5190 hospital admission events collected between January 2010 and October 2016 at Heart of England NHS Trust, Birmingham, UK (now part of University Hospitals Birmingham NHS Trust), which also includes a specialist cystic fibrosis (CF) unit. All patients from whose blood or urine cultures bacterial pathogens *Escherichia coli*, *Klesbiella pneumoniae*, or *Pseudomonas aeruginosa* was isolated were selected. Admission data (including Summary Hospital-level Mortality Indicator (SHMI) diagnostic codes, consultant specialties, admission methods, admission and discharge dates), patient demographics (age and sex), prescription records (also including antibiotics administered during any prior admissions dating from January 2010), and clinical records of culture tests (including antibiotic susceptibilities) were then searched for information linked to these patients. Since the cohort also includes a substantial number of inpatients from a specialist cystic fibrosis (CF) unit, we reported CF among comorbidities. Randomly generated patient identifiers and matching dates were used to link records. All categorical variables were stratified to binary variables for use within the machine learning algorithm. In total, 125 features were included as independent variables and tested for association with the outcomes; all the features considered are listed in Tables 1, 2, 3, and Tables A-B in S1 Text. These include patients' demographics, admission data, comorbidities, (historic and contemporary) antimicrobial drug prescriptions, and microbiology test results that determine whether the isolates are resistant to certain antimicrobial agents (Table 1). Resistance (R) and susceptibility (S) of isolates to four selected common antimicrobial agents (*co-amoxiclav* (AUG), *ciprofloxacin* (CIP), *meropenem* (MEM), and *piperacillin/tazobactam* (TAZ)) were also recorded as binary outcomes. Outcome predictions specific to each of these drugs were obtained from the subset of admissions linked to tests for susceptibility to each specific drug.

To perform predictions, we used gradient boosted decision tree (GBDT) models with logistic objective function implemented in the XGBoost library (v0.81) in Python (v3.7.1) [22, 23]. A GBDT aggregates a large number of weak models (decision trees) into a single prediction

**Table 1. List of antimicrobial agents considered in the study.** For each of these antibiotics and each admission event, binary variables record whether the antibiotic: a) was currently prescribed, b) has been prescribed to the same patient in a previous admission, c) resistant isolate was detected, d) susceptible isolate was detected, e) was prescribed within 72 hrs of admission (early prescription); other variables record f) the number of times resistant isolates were detected in previous admissions, d) the number of times susceptible isolates were detected in previous admissions. The 4th and 5th columns report the number of admissions which had resistance tested and outcome positive (R) and negative (S), respectively. Resistance during the current admission to the four drugs marked with a star (AUG, CIP, MEM, and TAZ) are outcomes set to be predicted by four different AI classifiers.

| Drug | Abbr. | Class | No. R | No. S |
|---|---|---|---|---|
| *Co-amoxiclav | AUG | Penicillin/$\beta$-lactamase inhibitor | 1455 | 2529 |
| *Ciprofloxacin | CIP | Fluoroquinolone | 1241 | 3883 |
| *Meropenem | MEM | Carbapenem | 532 | 4658 |
| *Piperacillin/tazobactam | TAZ | Penicillin/$\beta$-lactamase inhibitor | 1026 | 3376 |
| Amikacin | AK | Aminoglycoside | 384 | 1408 |
| Aztreonam | ATM | Monobactam | 644 | 134 |
| Ceftazidime | CAZ | Cephalosporin | 664 | 4065 |
| Cefotaxime | CTX | Cephalosporin | 367 | 3293 |
| Ertapenem | ETP | Carbapenem | 1259 | 3931 |
| Gentamicin | GT | Aminoglycoside | 745 | 4338 |
| Temocillin | TEM | $\beta$-lactamase-resistant penicillin | 103 | 2767 |

**Table 2. Pre-existing morbid conditions classified according to the SHMI diagnosis codes and abbreviations used throughout the text.**

| Abbreviation | Diagnosis description |
| --- | --- |
| Anemia | Deficiency and other anemia, Acute posthemorrhagic anemia |
| Arterial diseases | Aortic and peripheral arterial embolism or thrombosis |
| Atherosclerosis | Peripheral and visceral atherosclerosis |
| Bronchitis (acute) | Acute bronchitis |
| COPD/bronchiectasis | Chronic obstructive pulmonary disease and bronchiectasis |
| Cancer (rectum) | Cancer of rectum and anus |
| Cancer (secondary) | Secondary malignancies |
| Cancer (therapy) | Cancer; chemotherapy; radiotherapy |
| Cancer (uterus) | Cancer of uterus |
| Coronary diseases | Coronary atherosclerosis and other heart diseases |
| Enteritis/colitis | Regional enteritis and ulcerative colitis |
| Genital disorders (F) | Female genital disorders |
| Genital disorders (M) | Male genital disorders |
| Heart-valve disorders | Heart valve disorders |
| Hypertension | Essential hypertension, Hypertension with complications and secondary hypertension |
| Implant/graft | Complication of device, implant or graft |
| Infection (intestinal) | Intestinal infection |
| Infection (skin) | Skin and subcutaneous tissue infections |
| Infection (unspecified) | Bacterial infection; unspecified site |
| Lung disorders | Pleurisy; pneumothorax; pulmonary collapse |
| Lymphoma | Hodgkin's disease, Non-Hodgkin's lymphoma |
| Mental disorders | Mental retardation, Senility and organic mental disorders |
| Mycoses | Mycoses |
| Reprod. disorders (F) | Female reproductive disorders |
| Respiratory insuffic. | Respiratory failure; insufficiency; arrest (adult) |
| Septicaemia | Septicaemia, shock |
| White-cell diseases | Diseases of white blood cells |
| CF | Cystic fibrosis |

algorithm, where an individual tree consists of a series of nodes representing binary decision splits against one of the input variables (e.g., age>50 or history of antimicrobial resistance), with its final output being determined by the nodes at the end of the tree. GBDTs can robustly handle missing data and their predictions are virtually unaffected by multi-collinearity, thus often being more appropriate for healthcare data than alternative more traditional statistical methods such as logistic regression. A GBDT model depends on a number of hyper-parameters, which we selected by means of Bayesian optimisation [24] (Table C in S1 Text). The GBDTs were trained to predict the probability $P_i$ that a bacterial isolate is resistant to a selected agent is found during hospital episode $i$. As a measure of uncertainty of the prediction $P_i$ we consider the Gini impurity $G_i := 2 P_i (1 - P_i)$, which is zero when $P_i$ is either 0 or 1 (either susceptibility or resistant outcome is predicted with absolute certainty) and is maximum when $P_i = 0.5$ (both outcomes are equally likely).

Our main quantity of interest is the accuracy of both the recorded prescribing and of the GBDT predictions, key to this is the presence of mismatches between prescriptions/predictions and laboratory results. A prescription made by a physician is said mismatched if at least one bacterial isolate was found to be resistant to the drug administered, during an admission event. Assuming that a treatment can be administered only when the physician does not expect

**Table 3. Key covariates included in the GBDT classifier in addition to those of Table 2 and Tables A-B in S1 Text, and their abbreviations.**

| Abbreviation | Factor |
|---|---|
| Admi. date | Admission date |
| Age | Age in years |
| Early test | Laboratory test was performed within 72 hrs of admission |
| Frac. time in hosp. | Fraction of the time from first admission spent in hospital |
| LOS | Length of stay of current admission in days |
| M | Sex is male |
| # of admissions | Number of past admissions |
| # of comorbidities | Number of comorbidities |
| Total time in hosp. | Total time spent in hospital from the beginning of the study in days |
| ECOL | *E. coli* isolated in cultures |
| KPNE | *K. pneumoniae* isolated in cultures |
| PAER | *P. aeruginosa* isolated in cultures |

antimicrobial resistance (AMR), a mismatched treatment can be thought of a false negative (i.e., the physician predicted a negative AMR outcome while presence of resistant isolates was later confirmed by culture tests). On the other hand, we cannot know from the data if the physician predicted positive AMR outcome (obviously, the absence of treatment does not imply that the physician did not treat due to expected resistance). To allow fair comparison between the GBDT and the physician predictions, we then restrict our attention to the admissions where the drug was actually prescribed and contrast the false negative rates of the physician and GBDT predictions. This comparison requires extracting a binary prediction from the GBDT outcome probabilities (translating a probability into a Yes or No recommendation). In order to do so, we mark the $n$ admissions with the highest probability of resistance as positive to AMR (predicting that the bacteria is resistant); $n$ is taken as the true number of AMR outcomes in the set (chosen to maintain the true population-level percentage of outcomes). Then, we imagine treating only the inpatients who had predicted susceptible outcome *and* underwent true treatment, and tag these as AI assisted prescriptions. We finally compare the percentage of mismatches in the AI assisted prescription group with that in the true-treatment group. An example of ten different patient outcomes is given in Table 4, highlighting the AI decision steps and prescription mismatching. This strategy also lowers the total exposure to an antimicrobial agent. As a second mitigated strategy, we select the value of $n$ in such a way that the total number of matching AI-assisted prescriptions is the same as that of true matching prescriptions (strategy 2, Table D in S1 Text). In both strategies, the idea is to order all inpatients by their probability of AMR and administer the antimicrobial agent from the lowest to the highest probability until a condition is met. In other words, the AI algorithm would assist the drug prescription by skimming the group of inpatients deemed to receive antimicrobial treatment, excluding those that have a high probability of AMR. Throughout our analysis, Wilson intervals [25] are used as 2.5%-97.5% confidence intervals (CIs) for percentages.

To dissect the GBDT models and explain their predictions, we perform Shapley additive explanation (SHAP) analysis of the training dataset. We used an implementation specific to tree-based models, also referred to as TreeSHAP, accessible via the XGBoost and SHAP libraries [20, 21]. The Shapley values were introduced in game theory as a mathematical method for the allocation of credit among a group of players [19]. In the context of interpretable machine learning, these are optimal allocation of credit for the GBDT prediction among the $N = 125$ features included in the study, for each of the $M$ admission events. More specifically, each

**Table 4. Example of mismatched therapies and comparison between true and AI assisted prescriptions using strategy 1.** For a given antimicrobial drug and inpatient, data contains prescription records (first column—1 if the drug was prescribed during admission, 0 otherwise), and later culture test (sixth column—"True outcome", S or R if the isolate was found susceptible or resistant to the drug, respectively). In this example, 5 inpatients had resistant cultures, which defines $n$ in the algorithm. Drug treatment corresponds to a physician's prediction (second column—a physician prescribes a drug if they believe that isolates are susceptible to the drug, while nothing can be said if a drug is not prescribed). An AI system such as the GBDT model returns the probability that an isolate is found resistant (third column—"AI outcome"). The binary prediction is obtained by taking the $n$ = 5 cases with the highest probability and assigning a label accordingly (AI prediction—fourth column). This suggests not to administer a drug if resistance is predicted (AI prescription—fifth column), thus lowering the total exposure to the drug (from 6 to 4 inpatients under treatment). By comparing the true prescription or the AI prescription (columns 1 and 5) with the culture test results (column 6) the number of mismatches can be computed (7th and 8th columns for the physician and AI assisted prediction, respectively)

| True prescr. | Physician predic. | AI outcome | AI predic. | AI prescr. | True outcome | Physician mism. | AI mism. |
|---|---|---|---|---|---|---|---|
| 1 | S | 0.1 | S | 1 | S | 0 | 0 |
| 1 | S | 0.2 | S | 1 | S | 0 | 0 |
| 1 | S | 0.7 | R | 0 | R | 1 | 0 |
| 1 | S | 0.4 | S | 1 | S | 0 | 0 |
| 1 | S | 0.6 | R | 0 | R | 1 | 0 |
| 1 | S | 0.5 | S | 1 | R | 1 | 1 |
| 0 | ? | 0.1 | S | 0 | S | - | - |
| 0 | ? | 0.6 | R | 0 | S | - | - |
| 0 | ? | 0.8 | R | 0 | R | - | - |
| 0 | ? | 0.9 | R | 0 | R | - | - |

feature $j$, contributes a term $\phi_{ij}$ to the log-odds $f_i := \log(P_i/(1 - P_i))$ for episode $i$ (where $P_i$ is the probability of resistance from the GBDT algorithm). A negative value ($\phi_{ij} < 0$) implies that the feature $j$ contributes a negative term to the log-odds $f_i$ and thus has negative impact on the outcome, while a positive value ($\phi_{ij} > 0$) indicates a positive impact, for admission event $i$. The model output therefore satisfies $f_i = \sum_{j=0}^{N} \phi_{ij}$, where $\phi_{i0}$ is a bias term. Importantly, it has been mathematically proven that the Shapley allocation of $\phi_{ij}$ values is the only possible one that satisfies two additional desirable properties: firstly, if a feature's contribution increases or stays the same regardless of the other inputs, its Shapley value does not decrease (consistency property); and secondly a zero-valued feature contributes a zero Shapley value (missingness property). It is also possible to compute the Shapley values for a feature as differences between the predictions of the full model and the predictions of a model with that feature removed, averaged over all combinations of features held. In practice, there are far too many terms to evaluate this average exactly, but this provides an intuitive representation of the Shapley value as the additional predictive power enabled by the inclusion of each feature. For fast computation, we used an implementation specific to tree-based models, also referred to as TreeSHAP, available in the XGBoost and SHAP libraries [21]. By plotting the Shapley values for each admission event, it is possible to visually appreciate the scale and sign of each feature on the predicted outcome. Features can then be ranked by the absolute sum $\Sigma$ of their Shapley values (where for feature $j$, $\Sigma_j = \sum_{i=1}^{M} |\phi_{ij}|$ [21]), the slope index I ($I_j = \sum_{i=1}^{M} \phi_{ij} x_{ij} / \sum_{i=1}^{M} x_{ij}^2$, where $x_{ij}$ is the value of feature $j$ for admission $i$ [26]), and the information gain across all splits the feature is used in, averaged over all trees (see, e.g., [22, 27]). $\Sigma_j$ summarises the overall importance of feature $j$ for the prediction, regardless of whether the feature contributes a negative or positive term to the prediction; the slope index, I, provides a measure of association with a particular outcome; while the gain is a standard measure of feature importance available in the XGBoost library [23]).

The area under the curve of receiver operating characteristic (ROC-AUC or c-statistic with bootstrapped 2.5%-97.5% CIs) and the mean squared error (MSE) were used as appropriate measures of predictive power.

## Ethics statement

This was a retrospective analysis of a large volume of routinely collected clinical data. Individual patient consent was not obtained, but all information was pseudo-anonymised and ethical approval provided by the NHS Health Research Authority (reference 17/WM/0406).

## Results

### Varying confidence on predictions

The GBDT model provided drug-specific prediction of the probability of resistance, $P_i$, for each admission event $i$. Often in machine learning a threshold is used to translate this probability into a binary outcome—but the probability value contains important and useful insights. The receiver operating characteristic (ROC) curves comparing probabilities with binary outcomes are plotted in Fig A in S1 Text. The areas under the curve (ROC-AUCs or c-statistics) were 0.78 (95% CI 0.76–0.81), 0.87 (95% CI 0.85–0.89), 0.99 (95% CI 0.98–0.99), and 0.79 (95% CI 0.75–0.81), for resistance to AUG, CIP, MEM, and TAZ, respectively. These values are all relatively close to one, thus denoting excellent overall performances. Prediction of *meropenem* resistance is the most accurate, which may be due to repeated admissions of CF patients, for whom presence of AMR can be easily predicted. Given predictions of the probability, $P_i$, for each event and varying decision threshold $P^*$, such that cases with $P_i > P^*$ are predicted resistant, the receiver operating characteristic (ROC) curves show that many predicted outcomes match the true outcomes, while others inevitably do not, even for the high-AUC models. To help identify failures, the Gini impurity $G_i$ was computed for each prediction $P_i$, which can be thought of as a measure of the uncertainty of the prediction (Materials and methods). We separated the admissions into 10 groups based on their impurity ($0 \leq G_i < 0.05$, $0.05 \leq G_i < 0.1$, etc) and computed the mean-squared error (MSE) and the ROC-AUC in each group (shown in Fig 1 for the test set). Errors are low (close to zero) and ROC-AUC high (close to one) for the groups with the lowest impurity, that is for those predictions where the probability is close to one or zero. The greater predictive power for *meropenem* resistance is therefore directly attributable to a distribution of impurity scores dominated by low values (grey shaded region Fig 1), in comparison with the results for *co-amoxiclav* and *piperacillin/tazobactam*.

 This strongly suggests that the calculated probability (and hence the impurity score) provides clinicians with important information of the likely accuracy of the GBDT prediction. Predictions with low impurity are likely to be highly accurate, whereas those with high impurity are more subject to error—potentially due to a lack of information on which to base decisions.

### GBDT recommendations reduce mismatched treatments

To allow fair comparison of the GBDT predictions with medical staff decisions, we argue that for each event, all records corresponding to culture tests and drug prescriptions performed after the admission are discarded and we only include the information available at the time of admission. To this end, we removed all 'future information' from each admission event and fed the resulting set with missing data into the trained model. The outcome predictions (as captured by the area under the ROC curve) were virtually the same as the full model, scoring 0.77 (95% CI 0.74–0.80), 0.85 (95% CI 0.87–0.89), 0.99 (95% CI 0.98–0.99), and 0.78 (95% CI 0.75–0.81) ROC-AUC for predicting resistance to AUG, CIP, MEM, and TAZ, respectively. Discarding features recorded after admission has a limited impact on the predictions, thus showing that these features are largely redundant and have minor importance for the outcome. According to the global gain and $\Sigma$ metrics, the most important predictors were antibiotic

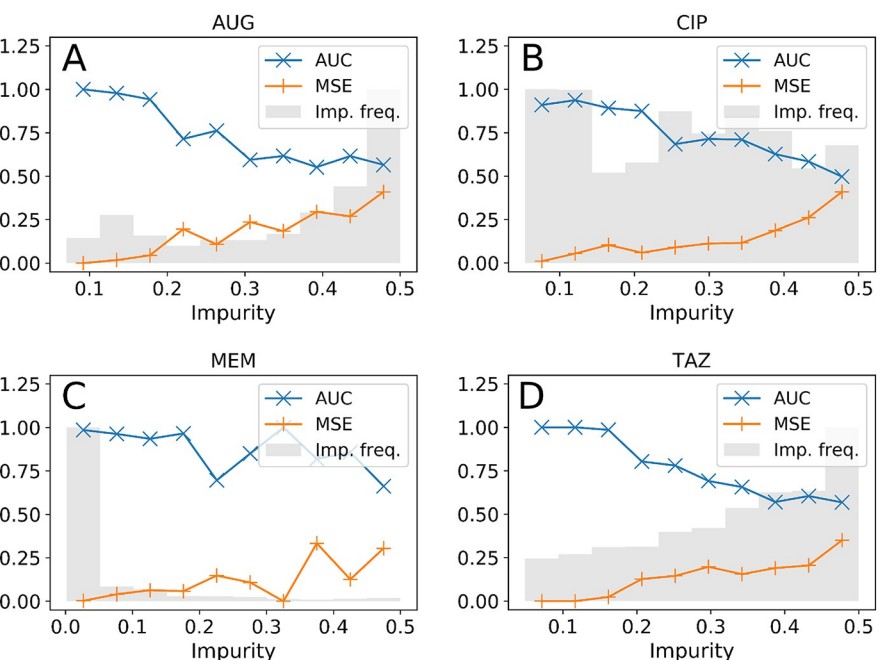

**Fig 1. Prediction accuracy vs prediction impurity.** Impurity frequency not to scale.

prescriptions and culture reports from past admissions, along with some past diagnoses (Table E in S1 Text).

We now turn to the question of whether GBDTs could be of practical health care benefit. Assisting prescriptions with GBDT predictions at the point of admission and before obtaining results on susceptibilities from laboratory tests, could have substantially reduced the percentage of mismatched treatment. To show this we restrict our attention to the subset of the inpatients who actually had the antimicrobial drug in question prescribed. As discussed in Materials and methods, these are individuals where the prescriber assumed the bacterial infection was susceptible to the drug; situations where a particular drug is not prescribed does not imply that the prescriber assumed the infection was resistant, other treatments may simply have been more appropriate. For these individuals we consider the GBDT prediction assigning resistance to a fraction of those with the highest probability (see Materials and methods). As a result, we find that the percentage of mismatches in this subset is substantially lower than in the set of true prescriptions (Table 5), although this comes at a cost of having to prescribe alternative drugs.

**Table 5. Mismatched therapies in true prescriptions (left columns) and prescriptions selected by the AI (center and right columns for strategy 1 and 2, respectively).** In these two strategies, the episodes most likely to report resistance to an antimicrobial agent are assumed to be left untreated (thus lowering the total exposure to the drug) or treated with a different drug. For all drugs considered, the percentage of mismatches is significantly lower with these two strategies than in true prescriptions. Mism. = mismatched.

| Drug | True prescriptions | | | AI assisted prescrip. (strategy 1) | | | AI assisted prescrip. (strategy 2) | | |
|---|---|---|---|---|---|---|---|---|---|
| | Mism. | Matches | % mism. | Mism. | Matches | % mism. | Mism. | Matches | % mism. |
| AUG | 631 | 1233 | 34% (32%–36%) | 210 | 1016 | 17% (15%–19%) | 557 | 1233 | 31% (29%–33%) |
| CIP | 53 | 209 | 20% (16%–26%) | 12 | 193 | 6% (3%–10%) | 40 | 209 | 16% (12%–21%) |
| MEM | 114 | 323 | 26% (22%–30%) | 7 | 307 | 2%(1%–5%) | 82 | 323 | 20% (16%–24%) |
| TAZ | 190 | 621 | 23% (21%–26%) | 91 | 560 | 14% (11%–17%) | 171 | 621 | 22% (19%–25%) |

In other words, by following this procedure, the clinician would have avoided treating with a certain drug those infections that were predicted, with high confidence, to be resistant to the drug. This choice would have lowered the mismatch percentage, and, at the same time, reduced the total usage of the antimicrobial agent.

### Factors differentially associated with AMR

We discuss here the associations between factors and predicted AMR outcome. For each admission event, the impact of the factors is defined by the Shapley values and the information encoded in the distribution of all Shapley values for a single factor is summarised by the slope index I (see Materials and methods). This represents a measure of association of the factor with the outcome and is positive when exposed admissions typically had positive impact (in Figs 2–5, S1–S5, S9 and S13 Figs red markers on the right side of zero—the values of I are reported on the left).

**Impact of antibiotic prescription.** We identified varying associations of resistance with past and current prescriptions. The Shapley values for any prescription during the admission event are summarised in Fig 2. This and the following figures condense a high volume of information into a single intuitive picture. From visual inspection of Fig 2-A (resistance to AUG,

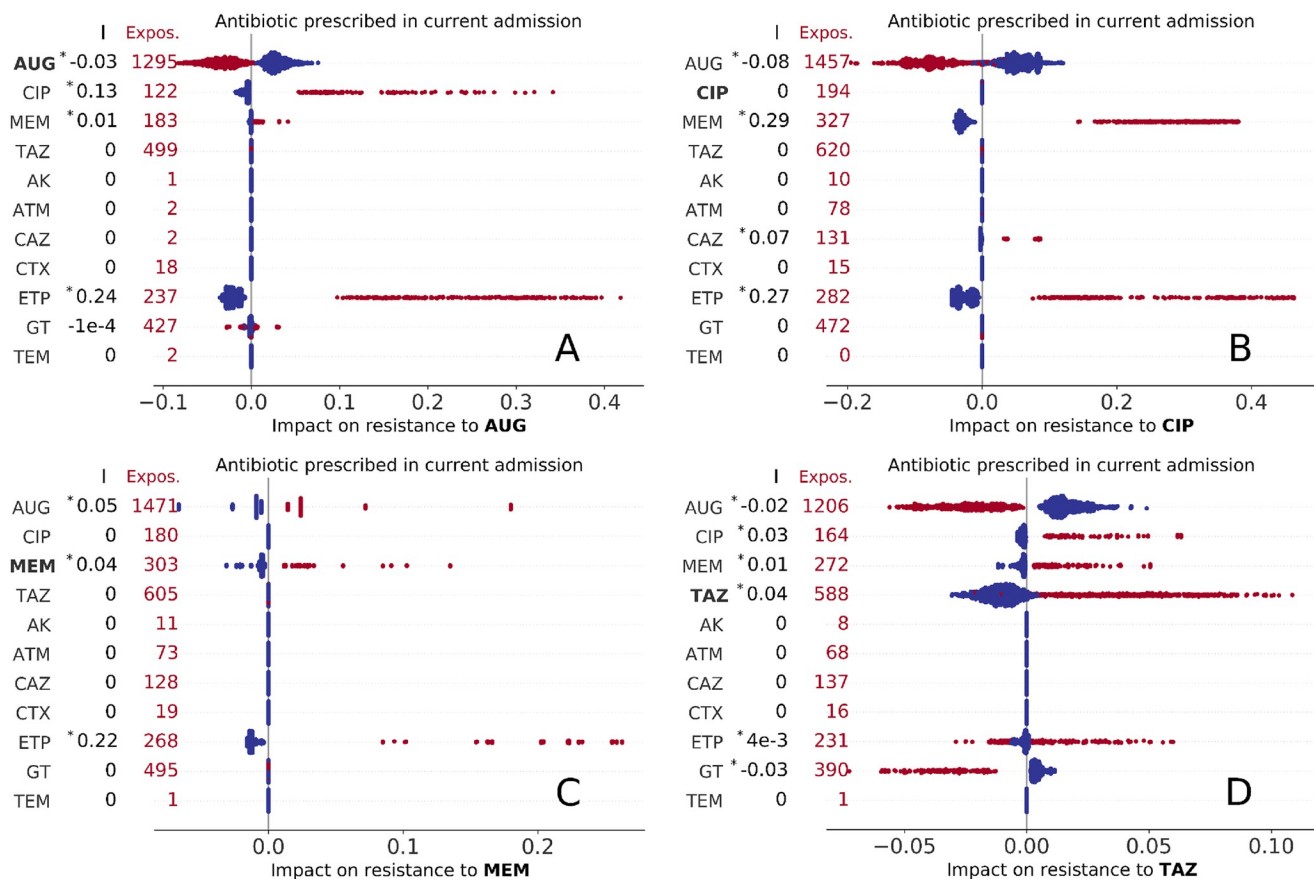

**Fig 2. Impact of present use of antibiotics on resistance to AUG, CIP, MEM, and TAZ treatment (A,B,C, and D panels, respectively).** Each marker represents an admission event, with colour red indicating exposure to the treatment and total number of treated episodes reported on the left for each drug. The corresponding Shapley values are represented as horizontal coordinates. Values of the slope index I are reported as measures of the direction and strength of association of the factor with AMR outcome (see also Table E in S1 Text, asterisk (*) indicates statistical significance, P < 0.01).

*co-amoxiclav*), it can be seen that admissions where CIP or ETP treatment occurred (represented by red markers in the second and third rows of scatter plots of Fig 2-A) had strong positive impact on predictions of resistance to AUG for all admissions (all corresponding Shapley values are positive, $\phi > 0$, with varying magnitude, $\phi$ ranging up to 0.4) and overall positive association (I = 0.13 and I = 0.24 for CIP and ETP, respectively). In contrast, admissions where AUG treatment is prescribed had negative impact ($-0.1 < \phi < 0$) on the prediction of resistance. Other treatments do not show such definite patterns of association, with the impact of use of GT being either negative or positive for different admission and typically much lower than other factors ($|\phi| < 0.1$, Shapley values smaller than 0.1 in absolute magnitude). In the next sub-plot (Fig 2-B), admissions where a *carbapenem* (MEM or ETP) is prescribed always had positive impact on resistance to CIP ($\phi > 0$, I = 0.29 and I = 0.27 for MEM and ETP, respectively), while AUG prescription had negative impact; CIP prescription did not impact resistance to CIP. Admissions with AUG, MEM, or ETP treatment had positive impact on resistance to MEM for all admissions (Fig 2-C), with ETP prescription having the highest association at I = 0.22. Finally, the impact on resistance to TAZ (Fig 2-D), was substantially lower in absolute values for all admissions ($|\phi| < 0.1$, I < 0.05), but being treated with TAZ had always positive impact, while AUG and GT negative impact). Interestingly, AUG was the only drug whose use negatively impacts prediction of resistance to the same drug—that is being prescribed AUG at any point during hospital stay is more likely to generate a prediction of susceptibility to AUG. We argue this might be due to AUG being prescribed only after AUG resistance was ruled out. It might be tempting to view the results in Fig 2 just as correlations between the observation and outcome. However, these Shapley values are more specific, showing how additional information on one particular characteristic (in this case the type of antibiotic prescribed during a stay in hospital) impacts the prediction for each single inpatient (in this case whether the infection is resistant). As such this is much more powerful than a simple correlation, extracting the additional predictive power that is gained from one additional feature. Of particular interest is the distribution of features (red and blue) markers across the range of Shapley values; in particular when there is a distinct split (that is red markers on one side of zero, blue markers on the other) the feature is a consistent sign of impact for all admissions.

Our analysis also included variables indicating whether an antimicrobial agent has been prescribed within 72 hours of admission (S1 Fig). Since the presence of resistant isolates is only revealed once laboratory test results have become available, we expect that early prescription of an antibiotic that matches with susceptibility to same drug reflects good stewardship. In fact, early prescription of AUG had a positive impact on resistance to AUG and TAZ ($\phi > 0$, I = 0.12 and I = 0.12, respectively, see top row of panels A and D in S1 Fig). This arguably follows from AUG being the first choice for treating infections before laboratory tests confirm their susceptibilities. Shapley values here are frequently small ($|\phi| < 0.1$), suggesting that the antibiotics prescribed in the first 72 hours are not overall a good predictor of resistance, with impact on resistance to broad-spectrum agents CIP and MEM essentially showing no trend (panels B and C in S1 Fig) We also observe that early prescription of MEM and CIP had negative association with resistance to AUG (I = −0.13) and TAZ (I = −0.27), respectively.

Of greater practical interest is the impact of historical antibiotic prescriptions (while in hospital) on predicted resistance. In general, prescription of any antibiotic is more likely to generate resistance, but there are some notable exceptions. In more detail, we find that a positive impact on resistance to AUG is observed for past prescription of AUG, CIP, TAZ, and GT (Fig 3-A). Interestingly, impact of past CTX usage was strongly negative for all exposed patients ($\phi < −0.2$, I = −0.28); this may suggests that AUG treatment might be a secure choice for patients previously treated with CTX. Past use of CIP had strong impact on resistance to the same drug

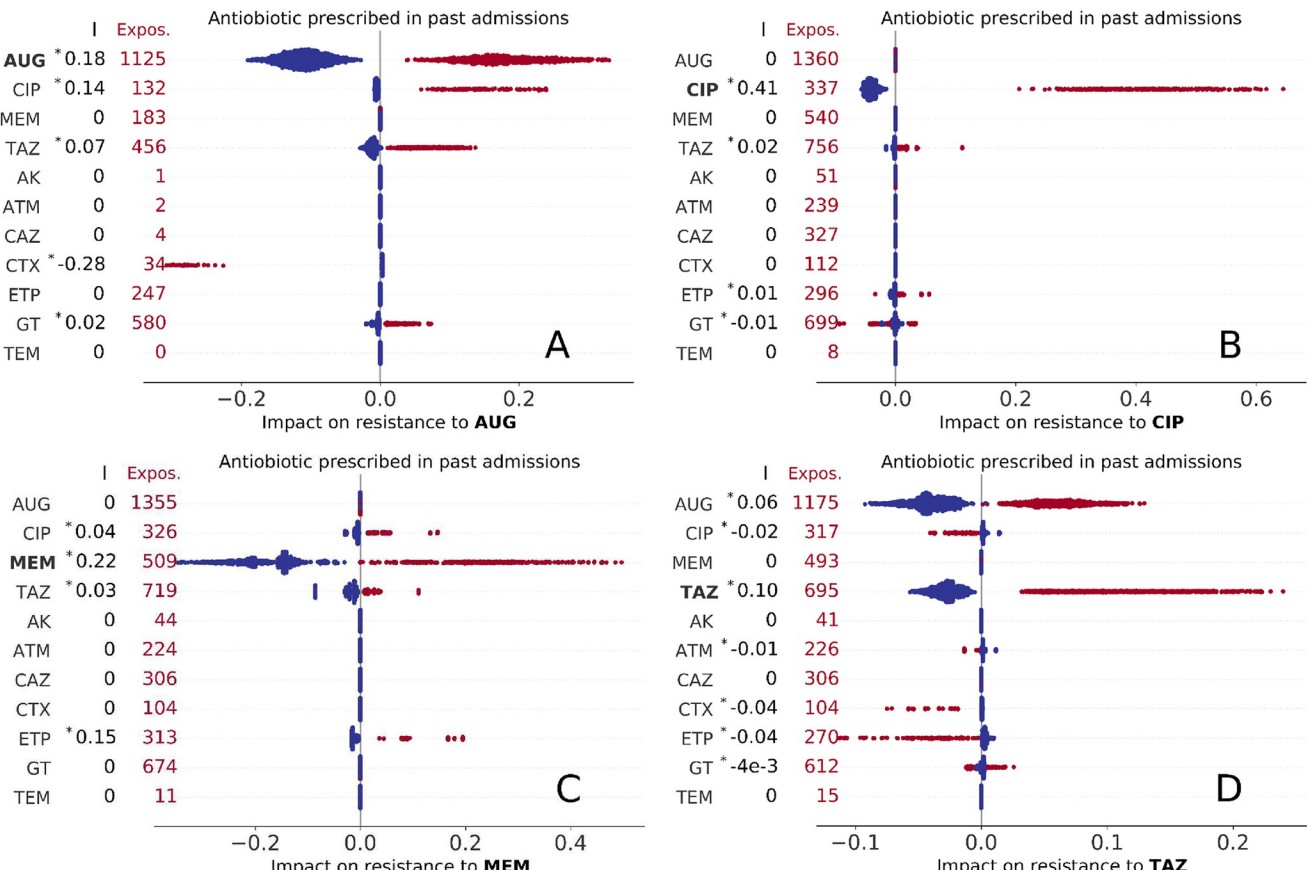

**Fig 3. Impact of past use of antibiotics (i.e., antibiotics prescribed in a previous admission) on resistance to AUG, CIP, MEM, and TAZ treatment (A, B,C, and D panels, respectively).** Each marker represents an admission event, with colour red indicating past exposure to drug treatment. The total number of episodes that were exposed to drug treatment in previous admissions are reported on the left for each drug. Keys as in Fig 2.

for all episodes (Fig 3-B, $0.2 < \phi < 0.7$, I = 0.41), while past use of other agents did not show any consistent association patterns. The patterns of past use of antibiotics on resistance to MEM are summarised in Fig 3-C; impact of past use of MEM or ETP was always positive (with indexes I = 0.22 and I = 0.15, respectively), while positive values were found for past use of CIP and TAZ with minor impact ($|\phi| < 0.1$). The impacts on resistance to TAZ were also minor (Fig 3-D), but with previous use of AUG and TAZ always yielding positive Shapely values. Impact of past use of a drug on resistance to the same drug was always positive.

**Impact of past resistance reports.** We now seek to identified impacts of previous culture records for the same-patient across different admissions—noting that both resistant and susceptible results may both confer useful information. For each admission and antimicrobial drug, we recorded the number of past admissions linked to resistant bacterial isolates and the number linked to susceptible isolates. The corresponding Shapley values are summarised in Fig 4 (resistance to the drug found in previous laboratory records) and S2 Fig (susceptibility previously found).

Overall, past detection of a resistance or susceptible isolate was positively associated with the presence of resistance or susceptibility to the same drug, respectively ($|\phi| > 1$ for all admissions and strong association of AUG with resistance to AUG, I = 0.58). This is by far the strongest impact on resistance result found, so past laboratory results are the single biggest

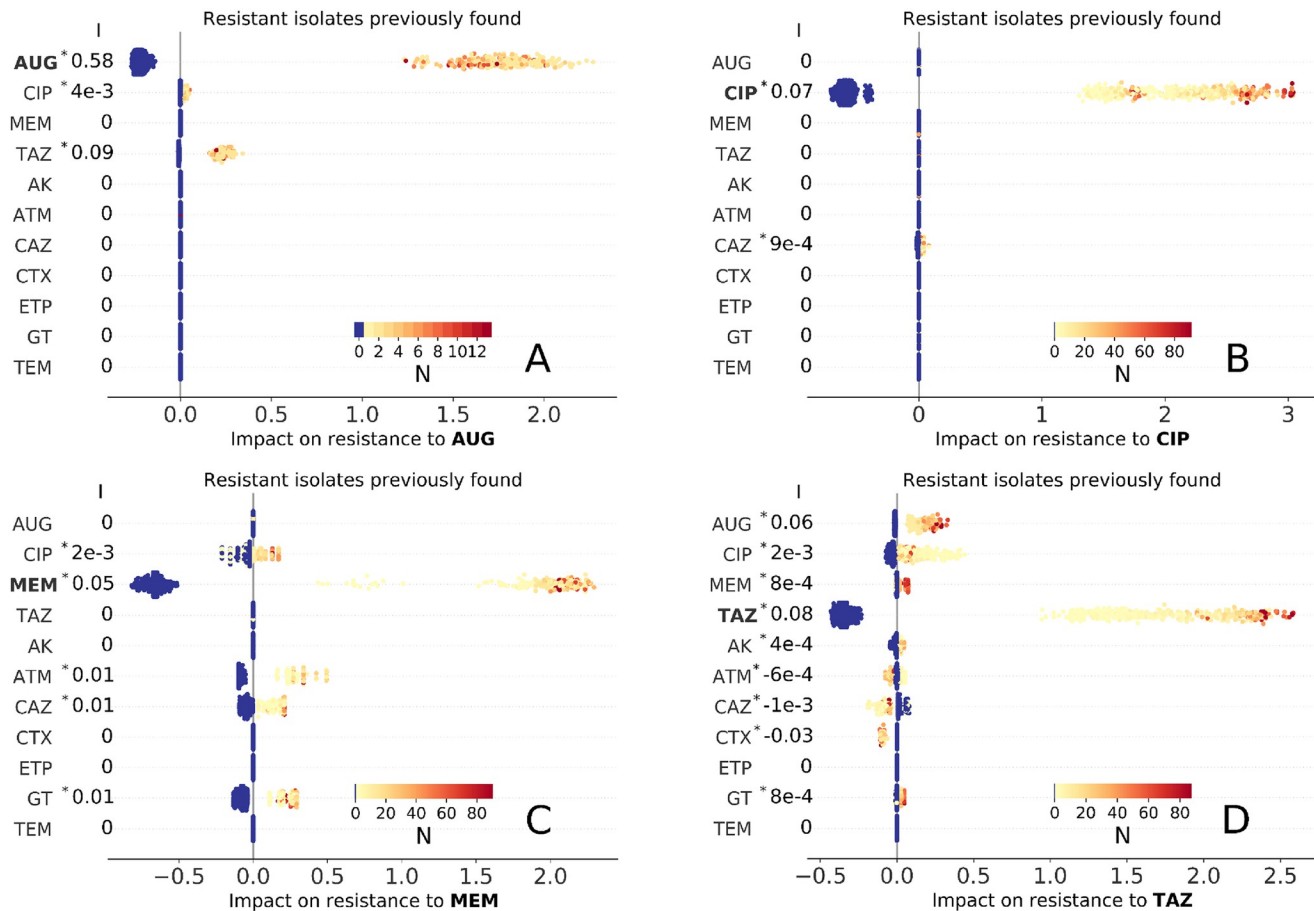

**Fig 4. Impact of past resistance reports (obtained by cultures tested during past admission) on resistance to AUG, CIP, MEM, and TAZ treatment (A,B,C, and D panels, respectively).** Blue marker color indicates no resistance previously found, while light orange to dark red colors correspond to increasing number N of times an isolate was found resistant in past admissions (inset colormaps). As in Fig 2, each marker represents an admission event and the corresponding Shapley values are represented as horizontal coordinates. Values of the slope index I are reported as measures of the direction and strength of association of the factor with AMR outcome (see also Table E in S1 Text, asterisk (*) indicates statistical significance, $P < 0.01$).

indicator of current resistance (see also Table E in S1 Text)—prescribing a drug where there had previously been resistance is a risky strategy. It is also worth noting the presence of cross-drug associations, especially those involving *co-amoxiclav* (AUG) and *piperacillin/tazobactam* (TAZ). Past resistance to TAZ and CIP had positive impact on resistance to AUG, even if with smaller magnitude than the impact from AUG (Fig 4-A); similarly, AUG and CIP had moderate positive impact on resistance to TAZ ($0.1 < \phi < 0.5$, Fig 4-D). Susceptiblity results are more subtle (S2 Fig). As expected, being susceptible to a given drug on a previous visit had a negative impact of being resistant to that drug on a future admission. There is again the cross-association between AUG and TAZ (panels A and D in S2 Fig). The importance of past resistance report is confirmed by the fact that by removing the corresponding variables, the overall accuracies of the GBDTs decrease substantially as quantified by their ROC-AUC metrics 0.63 (95% CI 0.60–0.65), 0.75 (95% CI 0.72–0.77), (0.97 (95% CI 0.96–0.97) and 0.59 (95% CI 0.55–0.62)) for AUG, CIP, MEM and TAZ respectively, which are smaller than the ROC-AUCs achieved using all predictors (albeit resistance to MEM is still accurately predicted).

**Impact of comorbidities.** Linking the dataset with SHMI diagnostic codes allowed us to find the impact of 29 morbid conditions on the predicted probability of AMR (Table 2).

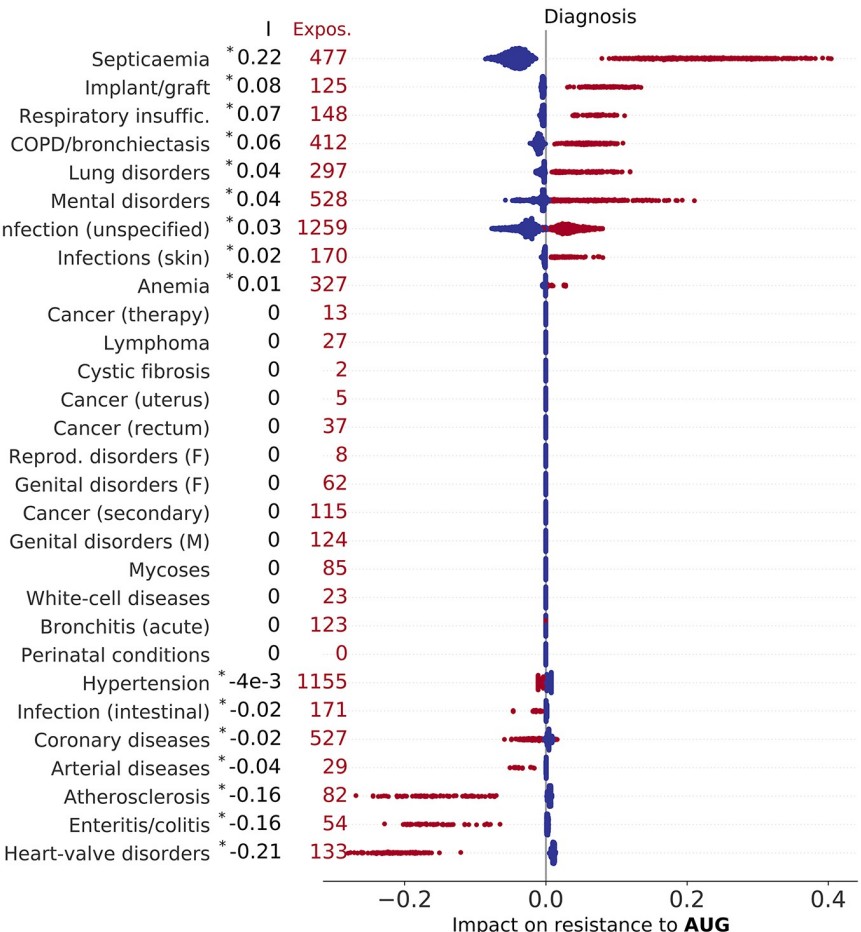

**Fig 5. Impact of morbidities (including cystic fibrosis and other diseases classified according to Summary Hospital-level Mortality Indicator (SHMI) codes) on resistance to AUG.** Factors are ranked by index I (top to bottom, values of I reported on the left), which measures the direction and strength of a factor's association with outcome (see also Table E in S1 Text, asterisk (*) indicates statistical significance, P < 0.01). Each marker represents an admission event, horizontal coordinates representing Shapley values, with colour red indicating presence of morbidity, and total number of diagnosed inpatients reported on the left for each disease (Expos.). The impact of diseases on resistance to CIP, MEM, and TAZ are illustrated in S3, S4 and S5 Figs, respectively.

Interestingly, many comorbidities are differentially associated with resistance to different antibiotics. The results are summarised in Fig 5 and S3, S4 and S5 Figs. Septicaemia has was the strongest risk factor for resistance to AUG among comorbidities (I = 0.22), cystic fibrosis had the strongest association with resistance to CIP and MEM (I = 0.54 and I = 0.59, respectively), while skin and subcutaneous tissue infection was strongest for resistance to TAZ (I = 0.22), consistently for all patients exposed ($0.1 < \phi < 0.8$).

An additional potential marker for resistance could be the total number of comorbidities, rather than the particular presence or absence of individual factors (S6 Fig). For CIP and TAZ the impact of the total number of comorbidities diagnosed was relatively low ($|\phi| < 0.1$). In contrast we observe a strong and distinctive pattern for AUG resistance (non-linear growing trend with saturation for AUG, impact always negative when the number of comorbidities is less than 4, panel A in S6 Fig); for MEM resistance (panel B in S6 Fig) there is only a substantial increase in risk when the number of comorbidities exceeds 11.

**Impact of age and sex.** S7 Fig illustrates non-linear relations with age. For all antibiotics except *ciprofloxacin*, the impact of age on resistance varied widely between the individual episodes and the antibiotics considered. For TAZ the age trend appears to increase in the old-patient group (age >80 years, panel D in S7 Fig) and a similar pattern, yet much less pronounced, can be observed for AUG (panel A in S7 Fig). On the contrary, for MEM the trend is overall decreasing, thus suggesting that the age was negatively associated with resistance to these agents, as illustrated in panel C in S7 Fig. It appears that the patterns of CIP, MEM, and TAZ might be driven by the presence of young CF patients. In particular, the Shapley values are positive for young ages, the age group with the highest prevalence of CF. Sex also had varying impact on antimicrobial resistance, albeit of relatively small magnitude, with male being positively associated with resistance to any drug except AUG (I = $-3 \times 10^{-3}$, 0.01, 0.09, 0.01 for AUG, CIP, MEM, TAZ, respectively, see S8 Fig).

**Impact of bacterial isolates.** The bacterial species isolated had contrasting associations with resistance to the five antibiotics considered. AUG is reported to be effective against clinical *Klebsiella* infections, but is not efficacious against *Pseudomonas* infections [28]; consistently, our Shapley value analysis highlighted negative impact of *K. pneumonia* infection for all episodes ($-0.9 < \phi < -0.3$ and I = $-0.62$), and zero impact for *P. Aeuruginosa* infections, due to the lack of exposure to AUG for *Pseudomonas* infections. Interestingly, the presence of *E. coli* also yielded zero impact (panel A in S9 Fig), such that the presence of *E. coli* infection does not change the prediction of resistance. The impacts of bacterial isolates on resistance to CIP and TAZ are low in magnitude compared to the other antimicrobial drugs (panels B and D in S9 Fig), albeit it is worth noting that the presence of *E. coli* always had positive impact for resistance to CIP. For resistance to MEM, the Shapley values are large for the impact of *E. coli* and *P. aeruginosa* (in absolute magnitude, $|\phi| > 0.5$), with *P. aeruginosa* yielding positive association (I = 0.94) and *E. coli* infection leading to negative association (I = $-1.52$), thus strongly suggesting that this class of drugs is more effective against *E. coli* than against *P. aeruginosa* (panel C in S9 Fig).

**Impact of other covariates.** Admission date had varying impact on AMR, particularly in terms of the drug considered (S10 Fig), reflecting changing levels of resistance within the population. The impact on resistance to AUG appears to have sharply decreased during the first year of the study, with further slightly decreasing impact over the remaining years (panel A in S10 Fig); a decreasing pattern can be observed also for CIP although of smaller magnitude (panel B in S10 Fig). Overall, this suggests that antimicrobial stewardship for these agents has been improving with time. On the contrary the impact on resistance to MEM and TAZ increased after 2013 (panels C and D in S10 Fig). In other words, being admitted on a later date had increased risk of resistance to MEM, which might indicate a potential emerging concern.

Duration of hospital stay can be measured in two ways: length of stay (LOS) for the admission event under examination (S11 Fig) and the total time a patient spent in hospital also including past admissions (S12 Fig). The LOS was generally a smaller risk factor than the total time a patient spent in hospital. With the exception of MEM, the longer was the total time spent in hospital, the higher was its impact on AMR. The growth appears logarithmic (horizontal axes are to logarithmic scale in S12 Fig), such that changes in the lower end are the most important for increased AMR risk. As a consequence, for CF patients (who tend to spend more time in hospital and populate the upper end of the range), changes in the time spent in hospital are in fact a less important factor for AMR compared to non-CF patients.

Some consultant specialties and admission methods also had an impact on AMR as illustrated in S13 Fig, arguably due to the fact that they carry some information on patients' health conditions.

### Patients' demographics

The total data set of hospital admissions includes 4507 individual patients (2760 male and 1747 female) with age at admission ranging from 18 to 105 years (median 78, inter-quartile range (IQR) 71 to 86). Median ages at admission of male and female patients were 75 and 80 years (IQRs 67–84 and 73–87), respectively.

## Discussion

Providing medical care is a complex process that can benefit from the support of AI models and algorithms. These can indeed outperform human predictions in speed and accuracy. However, mere prediction is only one aspect of healthcare. Even when optimised, probabilistic prediction always carries the risk that an AI decision system will be wrong and that, as a result, a patient might be injured. While mistakes in healthcare are always detrimental, involvement of AI might cause additional public mistrust and scale up a single algorithm error to injuries to many patients. Therefore, it is desirable that clinical support systems not only predict outcomes but also provide uncertainties and outputs interpretable to clinicians.

Here, we demonstrated that an appropriately designed AI can simultaneously achieve all of these goals. We worked with GBDT models trained to predict the presence of isolates resistant to *co-amoxiclav*, *ciprofloxacin*, *meropenem*, and *piperacillin/tazobactam* antimicrobial drugs as outcomes. Predicting probabilities of AMR (continuously ranging from no chances of resistance to virtual certainty of AMR, in contrast to just binary predictions) was a key factor. Assisting prescriptions with this information lowered the percentage of mismatches compared to real prescriptions. A measure of uncertainty was also associated with the probability and the error rate decreased with decreasing prediction uncertainty. The methodology can be used to help physicians reduce inappropriate use of antibiotic drugs (drugs unsuitable for a case being identified by high probability of AMR at admission) and select alternatives (drugs with lower probability of AMR, see also [15]). In particular, we recommend deploying broad-spectrum drugs (such as CIP and MEM) only when those with narrower spectrum are predicted to have high probability of resistance. This would prevent overuse of broad-spectrum agents, which is the cause of severe consequences including the emergence of antibiotic-resistant organisms [29].

Associations were identified based on the Shapley additive explanation, a technique of interpretable AI, summarising its comprehensive yet cumbersome output into so-called slope indexes I. Some of these associations are intuitive or expected, such as past use of a drug being associated with antimicrobial resistance to the same drug, while present use is negatively associated. Past detection of a resistance or susceptible isolate were also strongly associated with present resistance as suggested in other studies (see, e.g., reference [15]). This result suggests that the AI and the explanation models correctly weighted known risk factors of resistance. Data on past culture susceptibility are crucial and in their absence, the prediction accuracy as quantified by ROC-AUC metrics decreased significantly. Yet, in practise, these data may not be always readily available for all patients in need of care. This broadly poses a cautionary limit to the generalisability of data-driven predictions, which cannot be easily extended to different cohorts or naively applied in situations when crucial data is missing: validation studies are necessary. Our next steps also include attempts to model out situations where culture information or other variables are missing. The Shapley value analysis illustrates intriguing non-linear relations between resistance detection and age, admission date, and time spent in hospital. Septicemia and skin infection were the strongest risk factors among comorbidities for resistance to *co-amoxiclav* and *piperacillin/tazobactam*, respectively, while cystic fibrosis was highest for resistance to the broad-spectrum agents *ciprofloxacin* and *meropenem*. Sex male on average

appeared to have small positive impact on resistance, arguably to be attributed to unknown confounding factors such as past comorbidities or varying adherence to the prescribed regimen rather than to biological differences [30–32]. By identifying patterns and associations, the Shapley explanation has on one hand the potential to inform interventions and engage clinical personnel to improve stewardship. On the other hand—and more importantly—it can also be used by domain experts to evaluate complex automated clinical decision systems; by explaining how the AI weights factors to perform its prediction, the Shapley values can be used by physicians to determine whether AI recommendations are worthy of their trust.

As explainable AI demonstrates its capacity to summarise the effects of the predictive variables in complex data sets, we encourage the adoption of AI along with appropriate explanatory models in healthcare to support physician-led decisions and, in particular, to improve antibiotic stewardship.

## Supporting information

**S1 Text. Supporting information text, including: Fig A and Tables A, B, C, D, and E.** (PDF)

**S1 Fig. Impact of early use of antibiotics (i.e., antibiotics prescribed within the first 72 hours from admission) on resistance to AUG, CIP, MEM, and TAZ treatment (A,B,C, and D panels, respectively).** Each marker represents an admission event, with colour red indicating exposure to the treatment during the first 72 hours and total exposure reported on the left for each drug. Keys as in Fig 2. (PDF)

**S2 Fig. Impact of past resistance reports (obtained by cultures tested during past admission) on resistance to AUG, CIP, MEM, and TAZ treatment (A,B,C, and D panels, respectively).** Blue marker color indicates no susceptible isolates were previously found, while light orange to dark red colors correspond to increasing number N of times an isolate was found susceptible in past admissions (inset colormaps). All other keys are as in Fig 4. (PDF)

**S3 Fig. Impact of morbidities (including cystic fibrosis and other diseases classified according to Summary Hospital-level Mortality Indicator (SHMI) codes) on resistance to CIP.** Factors are ranked by index I (top to bottom, values of I reported on the left), which measures the direction and strength of a factor's association with outcome (see also Table E in S1 Text, asterisk (*) indicates statistical significance, P<0.01). Each marker represents an admission event, horizontal coordinates representing Shapley values, with colour red indicating presence of morbidity, and total number of diagnosed inpatients reported on the left for each disease (Expos.). (PNG)

**S4 Fig. Impact of morbidities (including cystic fibrosis and other diseases classified according to Summary Hospital-level Mortality Indicator (SHMI) codes) on resistance to MEM.** Keys as in S3 Fig. (PNG)

**S5 Fig. Impact of morbidities (including cystic fibrosis and other diseases classified according to Summary Hospital-level Mortality Indicator (SHMI) codes) on resistance to TAZ.** Keys as in S3 Fig. (PNG)

**S6 Fig. Impact of number of comorbidities on resistance to AUG, CIP, MEM, and TAZ (A, B, C, and D panels, respectively).** Red and blue markers correspond to cystic fibrosis (CF) and non-CF inpatients, respectively. Jitter along x axis introduced to avoid marker overlap.
(PDF)

**S7 Fig. Impact of age on resistance to AUG, CIP, MEM, and TAZ (A, B, C, and D panels, respectively).** Red and blue markers correspond to cystic fibrosis (CF) and non-CF inpatients, respectively. CF inpatients are younger on average (age<60) but the impact of age does not appear substantially different than non-CF.
(PDF)

**S8 Fig. Impact of other factors (Sex, early testing, fraction of days in hospital, number of hospital visits).** Exposures of male sex and early testing are (1008, 957), (1371, 1278), (1378, 1239), and (1255, 1082), for AUG, CIP, MEM, and TAZ, respectively.
(PDF)

**S9 Fig. Impact of *E. coli* (ECOL), *K. pneumonia* (KPNE), and *P. aeruginosa* (PAER) bacterial isolates on resistance to AUG, CIP, MEM, and TAZ treatment (A,B,C, and D panels, respectively).**
(PDF)

**S10 Fig. Impact of admission date (quantified by a Shapley value for each admission event, red and blue markers corresponding to cystic fibrosis (CF) and non-CF inpatients, respectively) on resistance to AUG, CIP, MEM, and TAZ (A, B, C, and D panels, respectively).** The impact varies non-linearly with the admission date.
(PDF)

**S11 Fig. Impact of length of stay (LOS) on resistance to AUG, CIP, MEM, and TAZ (A, B, C, and D panels, respectively).** Red and blue markers corresponding to cystic fibrosis (CF) and non-CF inpatients, respectively. Impact of LOS in CF inpatients is not substantially different than in non-CF inpatiens. Horizontal axis to logarithmic scale.
(PDF)

**S12 Fig. Impact of total time a patient stay in hospital including past admissions on resistance to AUG, CIP, MEM, and TAZ (A, B, C, and D panels, respectively).** Red and blue markers corresponding to cystic fibrosis (CF) and non-CF inpatients, respectively. CF inpatients appear to spent more time in hospital than non-CF, but the impact of time in hospital for these is typically lower (B and D, CIP and TAZ respectively). Horizontal axis to logarithmic scale and jitter introduced to avoid marker overlap.
(PDF)

**S13 Fig. Impact of consultant specialties to AUG, CIP, MEM, and TAZ.** Each marker represents an admission event, horizontal coordinates representing Shapley values. Colour red indicates if the consultant responsible for the care of the patient has specialty reported on the left. The total number of consultants of each specialty is also reported on the left (Expos.). Some consultant specialties might contain information on the patients' health and therefore also impact AI AMR prediction. According to the GBDT models, important predictors are "Infectious diseases" specialty, which appears to be associated with AMR to AUG (I = 0.22, $\Phi >$ 0.15), and "General medicine", which appears to be negatively associated with resistance to MEM (but positively associated to resistance to AUG and TAZ). Impact of consultant specialty on resistance to broad-spectrum agents CIP and MEM is often zero.
(PDF)

## Acknowledgments

We acknowledge support from the Warwick Bioinformatics RTP. We also thank Joht Chandan for valuable discussions.

## Author Contributions

**Conceptualization:** Massimo Cavallaro, Ed Moran, Christopher Green, Matt J. Keeling.

**Data curation:** Massimo Cavallaro, Ed Moran, Benjamin Collyer, Christopher Green.

**Formal analysis:** Massimo Cavallaro.

**Funding acquisition:** Ed Moran, Noel D. McCarthy, Christopher Green, Matt J. Keeling.

**Investigation:** Massimo Cavallaro.

**Methodology:** Massimo Cavallaro.

**Software:** Massimo Cavallaro, Benjamin Collyer.

**Validation:** Massimo Cavallaro.

**Visualization:** Massimo Cavallaro.

**Writing – original draft:** Massimo Cavallaro, Matt J. Keeling.

**Writing – review & editing:** Massimo Cavallaro, Ed Moran, Benjamin Collyer, Noel D. McCarthy, Matt J. Keeling.

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
