## [Decision Letter · Decision Letter 0]

4 Oct 2022

PDIG-D-22-00241

Informing antimicrobial stewardship with explainable AI

PLOS Digital Health

Dear Dr. Cavallaro,

Thank you for submitting your manuscript to PLOS Digital Health. After careful consideration, we feel that it has merit but does not fully meet PLOS Digital Health's publication criteria as it currently stands. Therefore, we invite you to submit a revised version of the manuscript that addresses the points raised during the review process.

Please submit your revised manuscript within 60 days Dec 03 2022 11:59PM. If you will need more time than this to complete your revisions, please reply to this message or contact the journal office at digitalhealth@plos.org. Please include the following items when submitting your revised manuscript:

We look forward to receiving your revised manuscript.

Kind regards,

Dukyong Yoon

Academic Editor

PLOS Digital Health

Journal Requirements:

a. Please clarify all sources of funding (financial or material support) for your study. List the grants (with grant number) or organizations (with url) that supported your study, including funding received from your institution. 

b. State the initials, alongside each funding source, of each author to receive each grant.

If you did not receive any funding for this study, please simply state: “The authors received no specific funding for this work."

2. We ask that a manuscript source file is provided at Revision. Please upload your manuscript file as a .doc, .docx, .rtf or .tex.

3. Please provide separate figure files in .tif or .eps format only and remove any figures embedded in your manuscript file. Please also ensure that all files are under our size limit of 10MB.

4. We have noticed that you have uploaded Supporting Information files, but you have not included a list of legends. Please add a full list of legends for your Supporting Information files after the references list. 

Additional Editor Comments (if provided):

The study design and results need to be modified in the clinical aspect. Please see the detailed comments from our reviewers.

Reviewers' comments:

Reviewer's Responses to Questions

**Comments to the Author**

1. Does this manuscript meet PLOS Digital Health’s publication criteria? Is the manuscript technically sound, and do the data support the conclusions? The manuscript must describe methodologically and ethically rigorous research with conclusions that are appropriately drawn based on the data presented.

Reviewer #1: Partly

Reviewer #2: Yes

2. Has the statistical analysis been performed appropriately and rigorously?

Reviewer #1: I don't know

Reviewer #2: Yes

3. Have the authors made all data underlying the findings in their manuscript fully available (please refer to the Data Availability Statement at the start of the manuscript PDF file)?

Reviewer #1: No

Reviewer #2: Yes

4. Is the manuscript presented in an intelligible fashion and written in standard English?

Reviewer #1: Yes

Reviewer #2: Yes

5. Review Comments to the Author

Reviewer #1: Summary

The authors showed that AI-based system in the present study can predict antimicrobial resistance and reduce unmatched prescription. Moreover, the authors emphasized that antimicrobial stewardship could be possible using a Shapley explanation model. This attempt was novel and challenging, but several concerns were found throughout the study procedure.

Major comment

1. Page 5, Line 77-78. The authors assumed that a prescription was done if a physician expect antimicrobial susceptible organism. On this basis, physician mismatch was determined. However, antimicrobial susceptibility is just a problem of what kind of antibiotics should be prescribed, not a problem of prescribing or not. When AI models predict the targets of antibiotic prescriptions, it is more appropriate to make decisions based on the distinction between infection and non-infection. 

2. The manuscript should be more concise. The content of Introduction, Methods, and Results were confused and mixed. The manuscript should be more classified each section according to their appropriate position. For example, the contents of Line 31-33 are more appropriate Methods section. 

3. The description of each variable that predicted antibiotic resistance (9-19 pages) needs to be presented more concisely. Only the main results should be presented in the Result section, and I recommend that the detailed results about factors associated with AMR could be moved to the supplementary. 

Minor comment

1. In the Methods section, please describe the statistical analysis method and the program used.

2. In the Result section, please provide baseline characteristics/demographics of the study population.

3. Page 2, Line 53. Among the data used as the input variable, consultant specialist is considered to be a variable unrelated to predicting antibiotic resistance. Why did the authors reflect this?

4. Page 20, Line 400-421. The second paragraph of the discussion and the first part of the third paragraph overlap in the Methods section.

Reviewer #2: Applications of machine learning to the medical field, including the early expert systems, continue to flourish. However, machine learning decisions are not always intuitive and are often criticized as a black box. In this study, the authors used information on patients’ demographics, data during hospitalization, and previous drug treatment to calculate the probability of resistance and predict the likelihood of antimicrobial resistance.

Although there have been many studies using artificial intelligence in the medical field, none have predicted drug resistance, and I believe this is a new study. The emergence of antimicrobial resistance has always been a topic of discussion since the development of antimicrobial agents, and in recent years, appropriate use of antimicrobial agents has been demanded, especially through action plans, but inappropriate administration of antimicrobial agents is still common. I hope that this study will promote the appropriate use of antimicrobial agents.

I believe that the following points need to be discussed.

1. The use of artificial intelligence generally requires a very large amount of data. For this reason, many studies have been conducted in high-income countries with large amounts of data, and it has been pointed out that this exacerbates healthcare disparities. The use of antimicrobial agents is often high in low-income and middle-income countries, and the generalizability of this study needs to be discussed.

2. I believe that artificial intelligence will not replace medical professionals, but will seek a way to provide medical care in cooperation with each other. It would be good to have a discussion on the interaction with medical professionals and how this research could be used in general medical practice.

3. Although culture information and prescribed medications in previous hospitalizations are listed, unlike patient background, these data may not always be available. In the absence of such information, how accurate are the predictions in this study? For example, if these data are not available, can the accuracy be maintained by using multiple imputation to fill in missing values?

4. Gender is assumed to have an impact on drug resistance, but it is well known that the disease trends are different between men and women, for example, women are more often affected by urinary tract infections. Could it be that past diseases are a confounding factor, resulting in the impact of gender?

5. Typo on line 279 "same same drug"

6. PLOS authors have the option to publish the peer review history of their article (what does this mean?). If published, this will include your full peer review and any attached files.

**Do you want your identity to be public for this peer review?** For information about this choice, including consent withdrawal, please see our Privacy Policy.

Reviewer #1: No

Reviewer #2: No

---

## [Decision Letter · Decision Letter 1]

14 Nov 2022

Informing antimicrobial stewardship with explainable AI

PDIG-D-22-00241R1

Dear Dr. Cavallaro,

We are pleased to inform you that your manuscript 'Informing antimicrobial stewardship with explainable AI' has been provisionally accepted for publication in PLOS Digital Health.

Best regards,

Dukyong Yoon

Academic Editor

PLOS Digital Health

Reviewer Comments (if any, and for reference):

Reviewer's Responses to Questions

**Comments to the Author**

1. If the authors have adequately addressed your comments raised in a previous round of review and you feel that this manuscript is now acceptable for publication, you may indicate that here to bypass the “Comments to the Author” section, enter your conflict of interest statement in the “Confidential to Editor” section, and submit your "Accept" recommendation.

Reviewer #1: All comments have been addressed

Reviewer #2: All comments have been addressed

2. Does this manuscript meet PLOS Digital Health’s publication criteria? Is the manuscript technically sound, and do the data support the conclusions? The manuscript must describe methodologically and ethically rigorous research with conclusions that are appropriately drawn based on the data presented.

Reviewer #1: Yes

Reviewer #2: Yes

3. Has the statistical analysis been performed appropriately and rigorously?

Reviewer #1: Yes

Reviewer #2: Yes

4. Have the authors made all data underlying the findings in their manuscript fully available (please refer to the Data Availability Statement at the start of the manuscript PDF file)?

Reviewer #1: Yes

Reviewer #2: Yes

5. Is the manuscript presented in an intelligible fashion and written in standard English?

Reviewer #1: Yes

Reviewer #2: Yes

6. Review Comments to the Author

Reviewer #1: Thank you for correcting the manuscript according to the reviewer's opinion.

I think the authors revised the manuscript more clearly and concisely.

Reviewer #2: The authors responded all the comments clearly and I'm satisfied with the revised manuscript.

7. PLOS authors have the option to publish the peer review history of their article (what does this mean?). If published, this will include your full peer review and any attached files.

**Do you want your identity to be public for this peer review?** For information about this choice, including consent withdrawal, please see our Privacy Policy.

Reviewer #1: **Yes: **Se Yoon Park

Reviewer #2: No
